# Risk of Falls Associated with Long-Acting Benzodiazepines or Tricyclic Antidepressants Use in Community-Dwelling Older Adults: A Nationwide Population-Based Case–Crossover Study

**DOI:** 10.3390/ijerph19148564

**Published:** 2022-07-13

**Authors:** Inyoung Na, Junyoung Seo, Eunjin Park, Jia Lee

**Affiliations:** 1Graduate School, Kyung Hee University, Seoul 02447, Korea; 2niy2@hanmail.net (I.N.); syrs929@hanmail.net (J.S.); bisang8112@naver.com (E.P.); 2College of Nursing Science, Kyung Hee University, Seoul 02447, Korea

**Keywords:** benzodiazepine, tricyclic antidepressant, adverse effect, falls, older adults

## Abstract

Background: Falls are common in older adults and increase in recent years. This study aimed to examine the risk of falls associated with long-acting benzodiazepines (BZDs) or tricyclic antidepressants (TCAs) use in community-dwelling older adults. Methods: A nationwide population-based case–crossover design was used. We screened information on 6,370,275 fall or fall fracture cases among community-dwelling elderly patients from the database of the national health insurance data warehouse in South Korea. We extracted the data of elderly patients who visited the hospital for a fall and were diagnosed with the first fall or fall fracture after prescription of long-acting BZDs (*n* = 1805) or TCAs (*n* = 554). The study used conditional logistic regression analysis to analyze the associations and stratified analysis by gender and age group to control for their confounding effects. Results: Risk of falls or fall fractures increased by more than two times after taking long-acting BZDs (odds ratio [OR] = 2.16; 95% confidence interval [CI] = 1.85–2.52) or TCAs (OR = 2.13; 95% CI = 1.62–2.83). The longer the prescription period of both, the higher the risk of falls or fall fractures was (≥49 days for long-acting BZDs vs. ≥ 56 days for TCAs). Conclusions: Long-acting BZDs or TCAs should be avoided or prescribed for a shorter duration based on these adverse effects. Health care providers should focus on fall prevention practices in older adults who take such drugs.

## 1. Introduction

Approximately 12.1–27.5% of older adults experience falls each year worldwide and there is a substantial increase in the number of falls in older adults in recent years [1,2,3]. Polypharmacy and high-risk medications have been reported to be the main cause of fall-related fractures [4]. Thus, it is necessary to be aware of the risk and the list of medications with negative biological and cognitive effects contributing to falls [5]. To address this issue, the Korea Health Insurance Review and Assessment Service (HIRAS) has created the Drug Utilization Review (DUR), a real-time alarm system, to educate medical staff on 20 geriatric high-risk drugs [6]. The system informs the high risk of adverse effects such as ataxia and orthostatic hypotension of 13 long-acting benzodiazepines (BZDs) and 7 tricyclic antidepressants (TCAs) [6]. Nevertheless, even after the introduction of the DUR system, drug use in older adults has not significantly decreased [7].

BZDs are the most commonly used sedatives for the treatment of sleep and anxiety disorders, and their effect suppresses the central nervous system more strongly than their sedative effect with increasing drug dosage [8]. In the most recent nationwide study, approximately 50.3% of Korean elderly outpatients had benzodiazepine prescriptions from January 2005 to June 2006 and 53.7% of them received long-acting formulations; 56.1% of patients who used these drugs experienced peptic ulcer and inflammatory disease and 23.2% experienced neurologic symptoms such as anxiety [9].

Furthermore, TCAs are also associated with increased adverse effects on the cardiovascular and autonomic nervous systems [10]. A Medicare beneficiary survey of community-dwelling older Americans also reported that the risk of falls among older adults increased by approximately 30%, especially among those who used antidepressants than among non-users [11].

Thus, health care providers should be conscientious of the high-risk medications and the consequences based on administration. This study aimed to analyze nationwide data and examine the adverse effects of long-acting BZDs and TCAs use on falls in community-dwelling older adults across all hospitals in South Korea.

## 2. Methods

### 2.1. Design

This study used a nationwide population-based case–crossover design using data from the national health insurance data warehouse (Figure 1). The case-crossover design is advantageous for transient exposures with short-term effects, particularly to reduce time-invariant between-subjects [12,13]. In this study, the time of fall or fall fracture was defined as the day of the diagnosis of falls or fall fractures in the outpatient clinics or emergency rooms of hospitals. We used 10 stages of the prescription period as comparison intervals (i.e., 7, 14, 21, 28, 35, 42, 49, 56, 90, and 120 days from the start of long-acting BZDs or TCAs before the first occurrence of a fall). We set the comparison intervals to up to 120 days because the Korea HIRAS recommends a gradual dosage reduction of 10–25% every 7–14 days for a period of up to 120 days depending on the patient’s condition [14].

### 2.2. Participants

Participant data were extracted from the database of the national health insurance data warehouse. The database includes the general characteristics of outpatients, their diagnoses, and medical treatments (e.g., procedure/surgery, medical examination, drug prescription, treatment materials, etc.) in all hospitals in South Korea. The inclusion criteria were: age ≥65 years, prescription with long-acting BZDs or TCAs, and diagnosis of a fall or fall fracture during a year. The exclusion criteria were: an additional diagnosis of a fall or fall fracture in the past year, a non-first fall or fall fracture after taking medications, and diagnosed with a fall or fall fracture for specific reasons such as slippery floors.

Among 6,370,275 fall or fall fracture cases in 2015, we identified total 1,427,968 older adults who took long-acting BZDs (*n* = 1,138,554) or TCAs (*n* = 289,414); after excluding 2003 older adults with an additional fall or fall fracture in 2014 for long-acting BZDs (*n* = 454) or TCAs (*n* = 1549), and 1,423,606 older adults who experienced a fall or fall fracture for specific causes such as slippery floors, total 2359 older adults who experienced the first fall or fall fracture after taking long-acting BZDs (*n* = 1805) or TCAs (*n* = 554) were included as study participants. Most were diagnosed with a fall fracture (Figure 1).

### 2.3. Measurements

#### 2.3.1. Falls or Fall Fractures

Fall or fall fracture diagnoses were measured using the Korean Standard Classification of Disease, 7th revision (KCD-7) based on the International Classification of Diseases, 10th revision (ICD-10) [15,16]. The KCD-7 includes 15 types of falls (i.e., ice and snow [W00], slipping, tripping and stumbling [W01], collision with, or pushing by [W03], while being carried or supported by [W04], wheelchair [W05], bed [W06], chair [W07], furniture [W08], playground equipment [W09], stairs and steps [W10], ladder [W11], scaffolding [W12], one level to another [W17], bumping against object [W18], Unspecified fall [W19]) and 11 types of fall fractures (i.e., skull and facial bones [S02], neck [S12], rib, sternum and thoracic spine [S22], lumbar spine and pelvis [S32], shoulder and upper arm [S42], forearm [S52], wrist and hand level [S62], femur [S72], Malleolus [S82], foot, except ankle [S92], multiple body regions [T02]).

#### 2.3.2. Long-Acting BZDs or TCAs Use

This was measured by the prescription records from the data warehouse and included the name of the drug, administration route, dosage, and duration of therapy. The long-acting BZDs included chlordiazepoxide (tablet), clidinium (tablet), clobazam (tablet), clonazepam (tablet), clorazepate (capsule), diazepam (tablet), diazepam (injection), ethyl loflazepate (tablet), flunitrazepam (tablet), flurazepam hydrochloride (tablet), mexazolam (tablet) and pinazepam (capsule). The TCAs included amitriptyline hydrochloride (tablet), amoxapine (tablet), clomipramine hydrochloride (capsule), imipramine (tablet), and nortriptyline (tablet) [7]. Quazepam from a long-acting BZD, and dothiepin hydrochloride and quinupramine from TCAs, were excluded because they were not covered under the National Health Insurance.

#### 2.3.3. Control Variables

Control variables were age and gender and were measured from the database of the national health insurance data warehouse.

### 2.4. Data Collection

Data on older adults who visited hospitals or clinics between January 2014 and May 2016 were extracted from the database of the national health insurance data warehouse after the approval of the Institutional Review Board (No. KHSIRB-17-058). The data extraction protocol was piloted with the records of 20 patients to determine their suitability. The data of patients’ characteristics, prescriptions, and diagnoses extracted from the database of the data warehouse were validated in a previous study [9] and widely used in national annual reports [17]. We invited five experts on big data analysis from the data warehouse to assess the content validity index (CVI) of the accuracy of data extraction. The CVI of the data extraction protocol was 0.92, indicating good content validity [18]. Based on their feedback, the extraction methods for the study variables were accepted. To prevent personal identification, the database converts patient data into an anonymized study ID. Moreover, we discarded all collected data after analyses and affirmed that these data were used solely for research purposes.

### 2.5. Statistics

Statistical analyses were performed using SAS, version 9.4 for Windows (SAS Institute, Inc., Cary, NC, USA). Data on participants’ characteristics were analyzed as the frequency, percentage, mean, and standard deviation. As this study used a case–crossover design that considers the effects of long-acting BZDs or TCAs, we conducted a conditional logistic regression analysis to analyze the association between the risk of falls or fall fractures and the use of long-acting BZDs and TCAs. We also performed stratified analysis by gender and age group to control for their effects. In the age-stratified analysis, the patients were stratified into five age groups: 65–69 years, 70–74 years, 75–79 years, 80–84 years, and ≥85 years. Statistical significance was set at a level of *p* < 0.05 for all analyses.

## 3. Results

### 3.1. Participant Characteristics

Among community-dwelling elderly patients who were prescribed long-acting BZDs or TCAs, the proportion of women was higher than that of men. Long-acting BZDs were mostly prescribed to patients with musculoskeletal and connective tissue diseases (*n* = 417; 23.1%), while TCAs were mostly prescribed to patients with mental and behavioral disorders (*n* = 223; 40.3%).

Most long-acting BZDs were prescribed for <30 days (*n* = 708; 39.2%), followed by ≥30 days and <90 days (*n* = 702; 38.9%). Most TCAs were prescribed for <30 days (*n* = 241; 43.5%), followed by ≥180 days (*n* = 124; 22.4%). In the number of prescription drugs, one pill was prescribed in 1,446 (80.2%) patients for long-acting BZDs and 486 (87.7%) patients for TCAs (Table 1).

### 3.2. Associations between Long-Acting BZDs or TCAs Use, and Falls or Fall Fractures

Conditional logistic regression analysis revealed a positive association between the risk of falls or fall fractures and the use of long-acting BZDs (odds ratio [OR] = 2.16; 95% confidence interval [CI] = 1.85–2.52; *p* < 0.001) and TCAs (OR = 2.13; 95% CI = 1.62–2.83; *p* < 0.001); thus, people who received these medications had twice the risk of falls or fall fractures than people who did not.

Regarding gender, the risk was twice more common or higher in male (OR = 2.76; 95% CI = 2.00–3.81) than in female (OR = 2.01; 95% CI = 1.69–2.39) who received long-acting BZD prescriptions (*p* < 0.001) compared to those who did not. There was also a positive association between the risk of falls or fall fractures and TCA in male (OR = 2.01; 95% CI = 1.15–3.55; *p* = 0.015) and female (OR = 2.17; 95% CI = 1.58–2.99; *p* < 0.001).

Regarding age, the highest risk of falls or fall fractures were among those 80–84 years of age (OR = 2.76; 95% CI = 1.99–3.83; *p* < 0.001) in long-acting BZD group; whereas in the TCA group more falls was seen in those ≥85 years of age (OR = 3.19; 95% CI = 1.41–7.22; *p* = 0.005).

In comparisons by prescription period, older adults taking long-acting BZD showed ≥twice the risk (OR = 2.02; 95% CI = 1.64–2.49; *p* < 0.001) of falls or fall fractures from 49 days or more after drug prescription, and the longer the period, the higher was the risk. Older adults prescribed TCAs showed twice the risk (OR = 2.00; 95% CI = 1.38–2.91; *p* < 0.001) of falls or fall fractures from 56 days or more after drug prescription, and the longer the prescription period, the higher was the risk (Table 2, Figure 2).

### 3.3. Risk of Falls or Fall Fractures Based on Prescription Period by Gender, and Age Group

Regarding prescription period by gender, 7 days after the prescription of long-acting BZDs, there was a positive association between the risk of falls or fall fractures and male (OR = 1.77; 95% CI = 1.05–2.99; *p* = 0.032) and female (OR = 1.39; 95% CI = 1.03–1.89; *p* = 0.033). Forty-nine days after the prescription of long-acting BZDs in male, the risk was ≥twice (OR = 2.44; 95% CI = 1.56–3.83; *p* < 0.001); similarly, 90 days after the prescription in female, the risk was ≥twice (OR = 2.59; 95% CI = 2.02–3.31; *p* < 0.001). Additionally, 49 days after the prescription of TCAs, the risk in female was ≥twice (OR = 2.30; 95% CI = 1.46–3.61; *p* < 0.001); 90 days after the prescription in male (OR = 2.70; 95% CI = 1.22–6.18; *p* = 0.014), the risk was ≥twice.

Regarding prescription period by age group, 7 days after the prescription of long-acting BZDs, the risk of fractures or falls was ≥ twice in the age group of 80–84 years (OR = 2.54; 95% CI = 1.34–4.82; *p* = 0.004); 42 days after the prescription, the risk was ≥twice in the age group of 70–74 years (OR = 2.07; 95% CI = 1.32–3.25; *p* = 0.002). Additionally, 42 days after the prescription of TCAs, people aged ≥85 years showed a 11-times-higher risk of experiencing falls or fall fractures (OR = 11.00; 95% CI = 1.42–85.20; *p* = 0.025). The risk was ≥twice in the age group of 75–80 years (OR = 2.56; 95% CI = 1.18–5.52; *p* = 0.017) (Table 3).

## 4. Discussion

This study investigated the risk of falls or fall fractures associated with long-acting BZDs or TCAs use through a nationwide population-based case-crossover design. In order to increase the accuracy of the results, this study analyzed the national data of community-dwelling elderly patients who visited a hospital for the first fall or fall fracture diagnosis after receiving the prescription drugs. The results showed that the risk increased by ≥twice after prescription of long-acting BZDs and TCAs in older Koreans, which is consistent with the finding of a previous study that analyzed the relationship between long-term BZDs use and other inappropriate psychotropics and the number of self-reported falls in older adults [19]. In our study, there were many older adults taking ≥90 days for long-acting BZDs (21.9%) and TCAs (34.7%) or taking two or more pills (19.8% vs. 12.2%). When comparing the overall risk, a longer duration of medications was increasing the risk of falls (≥49 days for long-acting BZDs vs. ≥56 days for TCAs). Moreover, when taken for ≥90 days, both medications showed 2.5 times and 2.6 times higher risk, respectively. Therefore, long-acting BZDs or TCAs should not be prescribed for ≥90 days, and drug management should be performed more thoroughly.

In our study, when both male and female older adults were prescribed long-acting BZDs, their risk at least doubled, and similar results were obtained when older adults of both genders were prescribed TCAs in a self-reported women study [20] and an inpatient fall study [21]. Moreover, 42 days after the prescription of long-acting BZDs, male older adults showed a risk that was 2.4 times higher; 90 days after the prescription, female older adults showed a 2.6 times higher risk. Additionally, 49 days after the prescription of TCAs, female older adults showed that the risk was 2.3 times higher; 90 days after the prescription, male older adults showed a 2.7 times higher risk. That is, in long-acting BZDs use, male older adults were at an increased risk of falls at an earlier period than female older adults while in TCAs use, female older adults increased the risk of falls at an earlier period than male older adults. A previous study also found that gender strongly affects the subjective effects and pharmacokinetic responses to drugs [22].

In our study, the risk was the highest when people aged 80–84 years were prescribed long-acting BZDs and when people aged ≥85 years were prescribed TCAs, indicating that the risk increased with age. Corroborating our results, previous studies showed that age is one of the greatest risk factors for fall fractures in older adults [23,24,25]. Therefore, for older adults (≥85 years of age), high-risk medication should be avoided or used cautiously, if at all possible, based on the adverse effects.

An analysis of risk based on prescription period by age group, in the age group of 80–84 years, 7 days after the prescription of long-acting BZDs showed that the risk was ≥twice. Moreover, 42 days after the prescription of TCAs, people aged ≥85 years showed an 11-times-higher risk of experiencing falls or fall fractures. Although the duration of medication should be evaluated based on the response and adverse effects, the use of TCAs and long-acting BZDs should be avoided in older adults (≥85 years of age) [26].

Based on our study results, we believe it necessary for health care providers to have more knowledge and collaborative practice regarding the need for long-acting BZDs and TCAs in older adult patients. If the nurse suspects a patient is high-risk, then they should communicate with other providers and maybe even decrease the dose or change the frequency. Patients should be carefully monitored, especially considering the risk of withdrawal syndromes, and those medications should be withdrawn safely [27]. Such knowledge may enable more well-informed decisions that improve care for community-dwelling older adults.

In older adults aged ≥65 years prescribed long-acting BZDs or TCAs, the prescription rate for females was approximately twice that for males. Moreover, a higher incidence of prescribing was not related to psychiatric conditions. The risk was ≥twice in patients taking geriatric high-risk drugs, and it increased with long-term use; if taken for ≥90 days, the risk was 2.5 times higher for long-acting BZDs and 2.6 times higher for TCAs. Thus, medical institutions are responsible for implementing a process for recognizing high-risk medications; and then the health care providers should identify patients who are at high-risk based on our study findings and develop management programs.

Our study had several limitations. First, the exclusive use of the database limited the ability to determine the actual use of the medications. Second, we analyzed 17 geriatric high-risk drugs, excluding three drugs, which were not covered under the national health insurance and for which the national health insurance data warehouse did not include information on. Third, this study was conducted only in South Korea; therefore, extending it to other cultures would be necessary. Fourth, as the focus was on older adults, study findings may not be extrapolated to younger adults, such as those aged 50 years.

## 5. Conclusions

The risk of falls or fall fractures can be reduced by ≥twice by limiting prescriptions for long-acting BZDs and TCAs in the elderly. The prescribers and the healthcare team who are directly caring for this population or their families should be informed of any risks associated with medications. Moreover, early assessment and identification are important to minimize and prevent long-term consequences. Based on the study findings, healthcare providers will be able to identify older adults at high risk of falls, such as those prescribed long-acting BZDs or TCAs, those taking geriatric high-risk drugs for ≥90 days, and those aged ≥80 years.

## Figures and Tables

**Figure 1 ijerph-19-08564-f001:**
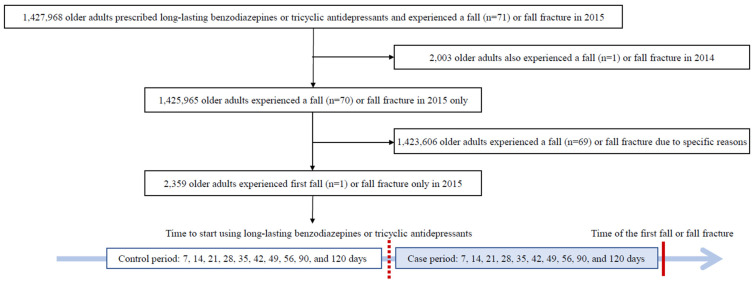
Flow chart of the case and control days selection by case-crossover design.

**Figure 2 ijerph-19-08564-f002:**
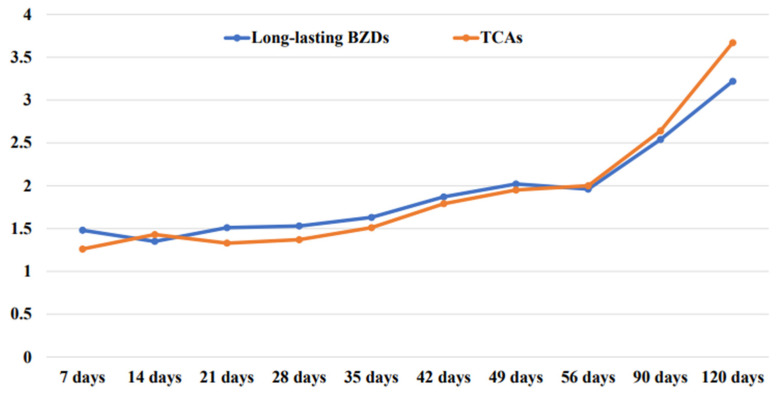
The relationship between the cumulative time of drug prescription and the increase in the risk of falls.

**Table 1 ijerph-19-08564-t001:** Participant Characteristics (*n* = 2359).

Characteristics		Long-Lasting BZDs (*n*, %)		TCAs (*n*, %)	
Subtotal (*n*)		1805		554	
Gender	Male	429	(23.8)	119	(21.5)
	Female	1376	(76.2)	435	(78.5)
Age group (years)	65–69	263	(14.6)	71	(12.8)
	70–74	436	(24.2)	133	(24.0)
	75–79	511	(28.3)	160	(28.9)
	80–84	371	(20.6)	118	(21.3)
	≥85	224	(12.4)	72	(13.0)
Principal diagnosis	Certain infectious and parasitic diseases	30	(1.7)	14	(2.5)
	Diseases of the circulatory system	157	(8.7)	43	(7.8)
	Diseases of the digestive system	264	(14.6)	26	(4.7)
	Diseases of the ear and mastoid process	152	(8.4)	2	(0.4)
	Diseases of the eye and adnexa	12	(0.7)	1	(0.2)
	Diseases of the genitourinary system	51	(2.8)	22	(4.0)
	Diseases of the musculoskeletal system and connective tissue	417	(23.1)	137	(24.7)
	Diseases of the nervous system	74	(4.1)	31	(5.6)
	Diseases of the respiratory system	110	(6.1)	11	(2.0)
	Diseases of the skin and subcutaneous tissue	26	(1.4)	2	(0.4)
	Endocrine, nutritional, and metabolic diseases	57	(3.2)	23	(4.2)
	Injury, poisoning, and other consequences of external causes	82	(4.5)	14	(2.5)
	Mental and behavioural disorders	319	(17.7)	223	(40.3)
	Symptoms, signs, and abnormal clinical and laboratory findings, not elsewhere classified	49	(2.7)	4	(0.7)
	Others	5	(0.3)	1	(0.2)
Prescription period (day)	1~29	708	(39.2)	241	(43.5)
	30~89	702	(38.9)	121	(21.8)
	90~179	167	(9.3)	68	(12.3)
	≥180	228	(12.6)	124	(22.4)
Number of prescribed pills per day (pill)	1	1446	(80.2)	486	(87.7)
	2	286	(15.8)	59	(10.6)
	3	53	(2.9)	8	(1.4)
	4	17	(0.9)	1	(0.2)
	≥5	3	(0.2)	-	-

**Table 2 ijerph-19-08564-t002:** Risk of Falls or Fall Fractures and Long-lasting BZDs or TCAs by Gender, Age, and Stage (*n* = 2359).

Variables		Risk of Falls or Fall Fractures, OR (95% CI), *p*	
		Long-Lasting BZDs (*n* = 1805)	TCAs (*n* = 554)
Subtotal		2.16 [1.85, 2.52], *p* < 0.001	2.13 [1.62, 2.82], *p* < 0.001
Gender	Male	2.76 [2.00, 3.81], *p* < 0.001	2.01 [1.15, 3.55], *p* = 0.015
	Female	2.01 [1.69, 2.39], *p* < 0.001	2.17 [1.58, 2.99], *p* < 0.001
Age group (years)	65~69	1.96 [1.30, 2.97], *p* = 0.001	2.44 [1.12, 5.28], *p* = 0.024
	70~74	2.17 [1.57, 2.98], *p* < 0.001	1.57 [0.82, 3.01], *p* = 0.174
	75~79	1.84 [1.38, 2.46], *p* < 0.001	1.97 [1.21, 3.19], *p* = 0.006
	80~84	2.76 [1.99, 3.83], *p* < 0.001	2.31 [1.30, 4.09], *p* = 0.004
	≥85	2.24 [1.45, 3.47], *p* < 0.001	3.19 [1.41, 7.22], *p* = 0.005
Prescription period (days)	7	1.48 [1.14, 1.93], *p* = 0.003	1.26 [0.76, 2.09], *p* = 0.371
	14	1.35 [1.09, 1.68], *p* = 0.007	1.43 [0.94, 2.18], *p* = 0.093
	21	1.51 [1.22, 1.86], *p* < 0.001	1.33 [0.89, 1.97], *p* = 0.163
	28	1.53 [1.24, 1.88], *p* < 0.001	1.37 [0.94, 2.00], *p* = 0.105
	35	1.63 [1.33, 2.00], *p* < 0.001	1.51 [1.05, 2.18], *p* = 0.028
	42	1.87 [1.52, 2.30], *p* < 0.001	1.79 [1.23, 2.60], *p*= 0.002
	49	2.02 [1.64, 2.49], *p* < 0.001	1.95 [1.35, 2.83], *p* < 0.001
	56	1.96 [1.60, 2.41], *p* < 0.001	2.00 [1.38, 2.91], *p* < 0.001
	90	2.54 [2.05, 3.14], *p* < 0.001	2.64 [1.83, 3.82], *p* < 0.001
	120	3.22 [2.58, 4.02], *p* < 0.001	3.67 [2.505.39], *p* < 0.001

OR, odds ratio; CI, confidence interval.

**Table 3 ijerph-19-08564-t003:** Risk of Falls or Fall Fractures from Long-lasting Benzodiazepines or Tricyclic antidepressants based on Prescription Period by Gender, and Age Group (*n* = 2359).

Variable			Risk of Falls or Fall Fractures, OR (95% CI), *p*									
			7 days	14 days	21 days	28 days	35 days	42 days	49 days	56 days	90 days	120 days
Gender	Long-lasting Benzodiazepine (*n* = 1805)	Male	1.77 [1.05, 2.99] *p* = 0.032	1.64 [1.03, 2.63] *p* = 0.038	1.68 [1.08, 2.62] *p* = 0.023	2.12 [1.32, 3.37] *p* = 0.002	1.93 [1.23, 3.02] *p* = 0.004	2.44 [1.56, 3.83] *p* < 0.001	2.65 [1.69, 4.17] *p* < 0.001	2.57 [1.66, 3.98] *p* < 0.001	2.40 [1.57, 3.67] *p* < 0.001	4.56 [2.74, 7.59] *p* < 0.001
		Female	1.39 [1.03, 1.89] *p* = 0.033	1.28 [1.00, 1.64] *p* = 0.053	1.47 [1.16, 1.86] *p* = 0.002	1.41 [1.11, 1.77] *p* = 0.004	1.56 [1.24, 1.96] *p* < 0.001	1.74 [1.38, 2.19] *p* < 0.001	1.87 [1.48, 2.36] *p* < 0.001	1.81 [1.43, 2.28] *p* < 0.001	2.59 [2.02, 3.31] *p* < 0.001	2.94 [2.30, 3.76] *p* < 0.001
	Tricyclic antidepressant (*n* = 554)	Male	2.00 [0.60, 6.64] *p* = 0.258	2.17 [0.82, 5.70] *p* = 0.117	1.75 [0.73, 4.17] *p* = 0.207	1.10 [0.47, 2.59] *p* = 0.827	1.40 [0.62, 3.15] *p* = 0.416	1.55 [0.72, 3.30] *p* = 0.261	1.33 [0.68, 2.60] *p* = 0.400	1.67 [0.82, 3.41] *p* = 0.162	2.70 [1.22, 6.18] *p* = 0.014	4.00 [1.74, 9.16] p = 0.001
		Female	1.13 [0.65, 1.98] *p* = 0.668	1.29 [0.81, 2.06] *p* = 0.287	1.23 [0.78, 1.92 *p* = 0.366	1.44 [0.94, 2.21] *p* = 0.090	1.54 [1.02, 2.33] *p* = 0.041	1.88 [1.22, 2.88] *p* = 0.004	2.30 [1.46, 3.61] *p* < 0.001	2.14 [1.38, 3.32] *p* = 0.001	2.61 [1.73, 3.95] *p* < 0.001	3.58 [2.32, 5.53] *p* < 0.001
Age group (years)	Long-lasting Benzodiazepine (*n* = 1805)	65~69	0.89 [0.45, 1.74] *p* = 0.732	1.50 [0.83, 2.72] *p* = 0.183	1.72 [0.96, 3.08] *p* = 0.067	1.72 [0.96, 3.08] *p* = 0.067	1.83 [1.03, 3.26] *p* = 0.039	1.62 [0.94, 2.79] *p* = 0.083	1.62 [0.94, 2.79] *p* = 0.083	1.85 [1.07, 3.19] *p* = 0.027	2.22 [1.27, 3.88] *p* = 0.005	3.80 [1.89, 7.63] *p* < 0.001
		70~74	1.55 [0.88, 2.72] *p* = 0.127	1.26 [0.79, 2.02] *p* = 0.340	1.79 [1.12, 2.84] *p* = 0.014	1.50 [0.96, 2.35] *p* = 0.076	1.66 [1.07, 2.57] *p* = 0.024	2.07 [1.32, 3.25] *p* = 0.002	2.40 [1.51, 3.83] *p* < 0.001	2.15 [1.36, 3.39] *p* = 0.001	2.42 [1.53, 3.83] *p* < 0.001	2.89 [1.88, 4.45] *p* < 0.001
		75~79	1.56 [0.96, 2.52] *p* = 0.073	1.17 [0.77, 1.76] *p* = 0.464	1.19 [0.77, 1.63] *p* = 0.564	1.50 [1.02, 2.22] *p* = 0.042	1.46 [1.00, 2.12] *p* = 0.050	1.52 [1.04, 2.23] *p* = 0.030	1.57 [1.08, 2.27] *p* = 0.018	1.54 [1.07, 2.22] *p* = 0.020	2.44 [1.64, 3.64] *P* < 0.001	2.82 [1.90, 4.19] *p* < 0.001
		80~84	2.54 [1.34, 4.82] *p* = 0.004	1.92 [1.18, 3.11] *p* = 0.008	1.86 [1.17, 2.94] *p* = 0.008	1.64 [1.06, 2.52] *p* = 0.026	1.87 [1.21, 2.90] *p* = 0.005	2.44 [1.56, 3.83] *p* < 0.001	2.79 [1.75, 4.45] *p* < 0.001	2.82 [1.73, 4.58] *p* < 0.001	2.52 [1.61, 3.93] *p* < 0.001	4.05 [2.45, 6.70] *p* < 0.001
		≥85	1.07 [0.53, 2.16] *p* = 0.858	1.08 [0.62, 1.89] *p* = 0.777	1.45 [0.82, 2.56] *p* = 0.201	1.32 [0.76, 2.30] *p* = 0.329	1.48 [0.87, 2.51] *p* = 0.148	1.85 [1.07, 3.19] *p* = 0.027	2.12 [1.19, 3.77] *p* = 0.011	1.90 [1.11, 3.27] *p* = 0.020	3.58 [1.89, 6.80] *p* < 0.001	3.31 [1.78, 6.15] *p* < 0.001
	Tricyclic antidepressant (*n* = 554)	65~69	0.80 [0.22, 2.98] *p* = 0.740	0.86 [0.29, 2.55] *p* = 0.782	1.00 [0.40, 2.52] *p* = 1.00	1.71 [0.68, 4.36] *p* = 0.257	1.86 [0.74, 4.66] *p* = 0.187	2.33 [0.90, 6.07] *p* = 0.083	2.00 [0.81, 4.96] *p* = 0.134	2.40 [0.85, 6.81] *p* = 0.100	3.50 [1.15, 10.63] *p* = 0.027	8.50 [1.96, 36.79] *p* = 0.004
		70~74	1.40 [0.44, 4.41] *p* = 0.566	2.40 [0.85, 6.81] *p* = 0.100	1.43 [0.54, 3.75] *p* = 0.470	1.10 [0.47, 2.59] *p* = 0.827	1.00 0.43, 2.31] *p* = 1.00	0.85 [0.38, 1.89] *p* = 0.683	1.09 [0.48, 2.47] *p* = 0.835	0.83 [0.36, 1.93] *p* = 0.670	2.00 [0.90, 4.45] *p* = 0.090	2.63 [1.16, 5.93] *p* = 0.020
		75~79	0.73 [0.29, 1.81] *p* = 0.493	0.81 [0.39, 1.69] *p* = 0.578	1.07 [0.53, 2.16] *p* = 0.858	1.19 [0.61, 2.31] *p* = 0.613	1.53 [0.80, 2.94] *p* = 0.198	2.56 [1.18, 5.52] *p* = 0.017	2.50 [1.20, 5.21] *p* = 0.014	2.36 [1.17, 4.78] *p* = 0.017	2.36 [1.26, 4.40] *p* = 0.007	4.00 [2.00, 8.00] *p* < 0.001
		80~84	2.50 [0.78, 7.97] *p* = 0.121	2.33 [0.90, 6.07] *p* = 0.082	2.00 [0.90, 4.45] *p* = 0.090	1.44 [0.62, 3.38] *p* = 0.396	1.36 [0.63, 2.97] *p* = 0.435	1.29 [0.64, 2.59] *p* = 0.481	1.54 [0.77, 3.10] *p* = 0.227	1.91 [0.92, 3.96] *p* = 0.082	4. 80 [1. 83, 12. 58] *p* = 0.001	2.89 [1.35, 6.17] *p* = 0.006
		≥85	2.50 [0.49, 12.89] *p* = 0.273	2.67 [0.71, 10.05] *p* = 0.147	1.33 [0.30, 5.96] *p* = 0.706	2.00 [0.60, 6.64] *p* = 0.258	3.00 [0.81, 11.08] *p* = 0.099	11.00 [1.42, 85.20] *p* = 0.017	11.00 [1.42, 85.00] *p* = 0.007	6. 50 [1.47, 28.80] *p* = 0.014	2.00 [0.81, 4.96] *p* = 0.134	4. 25 [1.43, 12.63] *p* = 0.009

## Data Availability

The datasets used and/or analyzed during the current study are available from the corresponding author on request.

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
