# Peer review of "Risk of Falls Associated with Long-Acting Benzodiazepines or Tricyclic Antidepressants Use in Community-Dwelling Older Adults: A Nationwide Population-Based Case–Crossover Study"

_ijerph, 2022, doi:10.3390/ijerph19148564_

Round 1
Reviewer 1 Report
The authors investigate that risk of falls associated with long-acting benzodiazepine of tricyclic antidepressants use in community-dwelling older adults: a nationwide population-based case-crossover study. The results demonstrated long-acting benzodiazepines or tricyclic antidepressants should be avoided in older adults and nurses should focus on fall prevention practices in older adults who take such drugs. It seems interesting, however, this article has some concerns to be addressed as follows.
The major concerns:
- This topic seems to be an old issue, that needs the innovated interventions rather than description of the relative risk of falls in association with specific study medications. In whole article, no significant suggestions in conclusions were given since this association is well-known and common policies to prevent the clinical risk were made in daily practicing.
- I just curious about the COVID-19 pandemics appears in the text, since there were any issue associated with the topic described in both ABSTRACT AND INTRODUCTIONS.
- In the study design, I didn’t see the study protocol that was written in METHODS, page 2, line 86. It’s very important to seen the protocol for further understanding the detailed study design.
- The authors recruited patients with falls or fall fracture, that should be clearly described how many population belongs to falls or fall fracture, respectively. Because the falls can result from no injury at all to severe adverse events that might be fatal. Therefore, more detailed data should be demonstrated.
Author Response
We truly appreciate your valuable comments. We made a table and used red font for the changes in the manuscript.

Reviewer 2 Report
Thank you the opportunity to review this interesting manuscript. The study is relevant however the authors fail to describe its importance and why it should be published among several other studies looking at inappropriate medicine use in older adults. Some specific comments below:
Abstract
- Authors state that: “Falls are common in older adults and increase during the COVID-19 pandemic.” The relevance and of covid and increased risk of falls is questionable here. Are the authors investigating this association?
- Methods: does not clearly explain what methods were used, e.g., were regression models used? Were there are confounders adjusted for?
- Results: “The longer the period, the higher was the risk of falls or fall fractures.” What period are the authors referring to?
- Conclusion: “Long-acting benzodiazepines or tricyclic antidepressants should be avoided in older adults because their adverse effects such as falls or fall fractures are serious.” This is a very strong recommendation considering these medicines are needed for various symptomatic management of chronic comorbidities. Please reword or revise, e.g. used with caution, etc.
Introduction
- Line 26: What data are the authors presenting, i.e., global prevalence/incidence of falls or local?
- Line 28-31 and line 31-35: very long and complicated sentences. This is evident throughout the document, particularly in multiple places where semicolon is used. Please simplify and reduce the use of semicolon.
- Line 48: Authors provide data for South Korea however there’s an insert of American data. How is this relevant to the current study?
Methods
- Control variables – are these confounders that were adjusted for? If so, then please mention.
Discussion
- As there are previous studies examining the same group of medicines and outcomes (e.g. reference no. 18, 19, 20, 21, etc), how is the current study different from those studies? What more does the current study add to existing literature?
- The discussion seems to repeat a lot of the results. I would encourage authors to cut short on the repetition and discuss how their results can make a difference to current practice. Is there a prescribing guideline to follow when using such medicines? If so, are they useful in practice?
- Line 53 “Thus, nurses should screen those drugs use in older…”. I am interested to know why this responsibility falls on the nurses? Aren’t prescribers educated and this education revised in practice? Isn’t the pharmacist involved in regulating medicine use?
- The authors have failed to discuss the strengths of their study and why should it be published in an already saturated pool of studies looking at inappropriate medicine use in older adults.
Author Response

(The authors gave the same response as above.)

Reviewer 3 Report
Dear Authors, although this paper touches on an interesting area of high scientific soundness and possible impact, there are some issues to be addressed:
- LINE 38: "Long-acting benzodiazepines are the most commonly used sedatives for the treatment of sleep and anxiety disorders" - please provide the citation and compare it with data on Z-drugs and short-acting ones like oxazepamum, lorazepamum, temazepamum.
- LINE 43: Can you provide some more novel data or present trends? This data seemed to be outdated.
- LINE 52: That's the conclusion, not appropriate to summarize the introduction section.
- LINE 76: The exclusion criteria are really short-listed. Please provide full detailed exclusion criteria to the study protocol.
- LINE 91: Please provide the comparison between ICD-KCD (Table?) that will allow the peers to catch up "on the same page" of diagnostic criteria.
- Page 9, LINE 2: This sentence may be deleted.
- Discussion section: the part I can see which is missing is the benzodiazepine withdrawal risk - connected to the low compliance with patients. Please analyze your data and provide a discussion on the risk of falls connected to seizures or acute withdrawal, and hypotension due to tricyclics use. The discussion on the registries and drug use and possible withdrawal and the Drug Utilization Review may be enriched in papers touching similar subjects, such as doi: 10.12740/PP/115555 or doi: 10.1007/s11920-016-0727-9 .
- Those above are to be mentioned also in the limitations of the study.
Author Response

(The authors gave the same response as above.)

Reviewer 4 Report
I appreciate the hard work on the manuscript, but hope you will consider the suggestions and comments for improving the content and eliminating many repetitive sections/words.

Author Response

(The authors gave the same response as above.)

Round 2
Reviewer 1 Report
Accept
Author Response
Thank you for your thoughtful review and wonderful decision.
Reviewer 3 Report
Although not every one of my reported issues was addressed, I believe that even if there is a lack of novelty in such studies, once they are published in a serious journal, they play an important role to awaken the consciousness in not only the studied country, and may contribute to changes in public health policies.
To the authors. I was reading carefully your answers to other reviewers. My recommendation is to remain calm and never confront the reviewer with offensive: "!" signs. This is not a chat discussion, where you can say: "Yes, it was already deleted!". When the reviewer suggests deleting some of the not important sentences that are not deleted in the revisioned version - support it and explain.
Author Response
We appreciate your valuable recommendation.
For all future manuscripts, we will add more logical explanations to the reviewer's comments.
Thank you for your thoughtful review and wonderful decision.